# The Research of Standardized Protocols for Dog Involvement in Animal-Assisted Therapy: A Systematic Review

**DOI:** 10.3390/ani11092576

**Published:** 2021-09-02

**Authors:** Antonio Santaniello, Susanne Garzillo, Serena Cristiano, Alessandro Fioretti, Lucia Francesca Menna

**Affiliations:** 1Department of Veterinary Medicine and Animal Productions, University of Naples Federico II, 80134 Naples, Italy; susannegarzillo@gmail.com (S.G.); alessandro.fioretti@unina.it (A.F.); 2SInAPSi Center, University of Naples Federico II, 80133 Naples, Italy; serenella25@live.com

**Keywords:** animal-assisted therapy (AAT), dog therapy, public health, choice of co-therapist dog, health protocols, dog welfare, one health

## Abstract

**Simple Summary:**

Animal-assisted therapies (AATs), as discussed in this review, are structured interventions, involving pets, for patients suffering from different diseases. Although many studies have highlighted the beneficial effects of these interventions on the well-being and health of the humans given the dogs, there are few studies that highlight the involved dogs themselves. Therefore, in this study, we carried out a systematic review to investigate the characteristics of the dogs involved in AATs. Based on the results, in most papers, there is a lack of general information on the dog(s) involved, including the methods used to choose and train the dog and the animal’s health status. These results highlight the need for standardized, specific methods to choose and train the dog and also suggest the need for univocal health protocols to ensure the animal’s welfare, as well as the final results of the therapeutic intervention.

**Abstract:**

Dogs are considered the most important species involved in animal-assisted therapy (AAT), and the scientific literature focuses on the benefits linked to the involvement of dogs in various therapeutic areas. In this study, we carried out a systematic review according to the Preferred Reporting Items for Systematic Reviews and Meta-Analyses (PRISMA) guidelines, exploring the scientific literature from the last 5 years (2016–2021) on three databases (PubMed, Scopus, and Web of Science) to highlight the characteristics of the dogs involved in AATs. Based on the scientific literature relevant to such dogs, we considered different parameters (i.e., number, age, sex, breed, temperament, methods of choice and training, health status, research goals, and activities with dogs) to include studies in our paper. After screening 4331 papers identified on the searched databases, we selected 38 articles that met the inclusion criteria. Analysis of the included articles showed that the characteristics of the dogs were neglected. Our findings indicated a lack of information about the dogs, as well as the absence of standardized and univocal criteria for dog selection, training programs, and health protocols.

## 1. Introduction

According to some archaeological evidence, the domestication of dogs took place more than 30,000 years ago and was fundamental for the evolution of humans [1]. The dog was the first species to be domesticated. This domestication process was born from an extraordinary event in which suitable ecological conditions allowed the coexistence and, therefore, the formation of a relationship between human and dog. The coevolution between human and dog has created a unique interspecific relationship that has spread across cultures around the world and has survived by changing over the years [2]. Dogs were selected to cooperate with humans through domestication and, thus, evolved some genetic predispositions that allowed them to develop skills shared with humans [3,4,5], representing a case of convergent evolution between two species [6]. Dogs have various interspecific abilities due to their ability to read behavior and understand human communication. Dogs pay attention to social signals, can respond appropriately to human facial expressions [7], and are able to look at a human face and follow that person’s gaze [8]. In particular, dogs can extract and integrate emotional information and discriminate between positive and negative human emotions [9], as well as recognize the vocalizations of emotions felt by people [10]. One of the greatest skills of dogs is their ability to read the non-verbal language of humans [11]. This strong interspecific bond has been demonstrated by studies on attachment between dogs and humans. The close behavioral correspondence between dogs and human is widely recognized and gives the animals the necessary fundamental bonds of attachment to form a relationship [12,13]. These social skills have made it possible to involve dogs as valid co-therapists in AAT. Indeed, the dog is the main species involved in the scientific literature on AAT [14,15,16].

As reported by the International Association of Human-Animal Interaction (IAHAIO) White paper [17] regarding the field of animal-assisted interventions (IAAs), AATs are structured therapeutic interventions directed and/or provided by health, education, and/or human service professionals [17].

AATs represent an adequate expression of integrated medicine according to the One-Health approach. According to the 12 Manhattan Principles, recognizing the link between human beings and animals will help safeguard and improve the health of the community. Using this approach, we consider a zooanthropological vision of healthcare that studies and applies an interspecific relationship between humans and animals in both healthcare/therapeutic and educational contexts.

Current theories concerning the mechanisms responsible for the therapeutic benefits of AATs agree on the idea that animals possess unique qualities that can facilitate and contribute to the success of a therapy [18]. Furthermore, developing a relationship with an animal in therapy can lead to positive changes in cognition and behavior through the acquisition of new skills, as well as the acceptance of personal actions and responsibilities [16,19]. The current research regarding the effects of AATs has primarily addressed human health outcomes. The therapeutic benefits are both physiological (i.e., a reduction in stress, heart rate, etc.) and psychological (i.e., treatment of post-traumatic stress) [20]. Most studies focused on the benefits that a therapy dog can provide in different areas, such as autism spectrum disorder [21,22], stress [23,24], Alzheimer’s disease [25], the school environment [26,27], rehabilitation [28,29], the hospital environment [30,31], and Post-Traumatic Stress Disorder (PTSD) [32,33].

As the dog is the focus of AAT interventions, scientific studies should not ignore, in particular in materials and methods, defining and/or declaring the characteristics of the dog involved (age, sex, breed, temperament, tests for the choice, health, and behavioural check), since they influence the progress and the effects of such interventions [15,16,18,19,20]. Therefore, the aim of this systematic review was to outline the characteristics of dogs involved in AAT by highlighting the information reported by relevant scientific literature over the last 5 years.

## 2. Materials and Methods

This review was performed according to the Preferred Reporting Items for Systematic Reviews and Meta-Analyses (PRISMA) [34] using the following steps: (1) preparation of a database search to detect potentially related articles, (2) assessment of the relevance of papers, (3) the evaluation of eligibility, and (4) data extraction.

### 2.1. Inclusion and Exclusion Criteria

In this Systematic Review, although we also explored the possibility of a grey-literature search, only original English studies (published or in press) were included, while reviews, meta-analyses, retrospective studies, case reports, commentaries, notes, and letters to the editor were excluded.

Moreover, only papers published during the last 5 years (from 2016 until 31 May 2021) were selected to ensure a more complete overview of the recent scientific literature.

Only articles reporting on therapeutic interventions with dogs (AAT) were included, while studies concerning other types of AAIs (e.g., animal-assisted activity and animal-assisted education) were excluded.

### 2.2. Search Strategy and Data Sources

In our study, published scientific literature was selected by searching three electronic scientific databases: PubMed [35], Scopus [36], and Web of Science [37].

The systematic search was performed by two researchers (S.G. and S.C.) independently using the following strings: “Animal-Assisted Therapy” and “dog”, “Dog-therapy”, “Dog-assisted therapy”, “Canine-assisted therapy”, and “Canine-assisted psychotherapy”. All terms were selected based on international reference guidelines [17,38].

Other related papers, including references from the selected papers, were revised and used as sources of supplementary information.

### 2.3. Study Selection and Data Extraction

The relevant information was extracted from each paper included in this systematic review to achieve our stated goal. All data were entered into an Excel data set. We included data related to the characteristics of the dogs (i.e., number, age, sex, and breed), whether the dog was handled or owned, methods for choosing the dog, the temperament and training of the dog, and the health and behavioral status of the dog. Additional data were extracted to facilitate identification of each study (i.e., first name and year of publication).

The search query identified 114 articles (69 in PubMed, 32 in Scopus, and 13 on the Web of Science). After evaluating the titles and abstracts of all articles, we selected the papers; after removing irrelevant and duplicate papers, only papers published in journals indexed on the Pubmed [35], Scopus [36], and Web of Science [37] were included. Ultimately, a total of 38 papers met the inclusion criteria. Figure 1 illustrates the PRISMA (Preferred Reporting Items for Systematic Reviews and Meta-Analyses) flow-chart process [34] for study selection.

The inter-judge agreement was calculated (and independently identified by two judges) as a measure of reliability and assessed by Cohen’s kappa. The minimum reliability value was ≥0.81, indicating strong agreement between the judges. Every disagreement was solved through intervention of the senior author (A.S.).

## 3. Results

The preliminary database search returned a total of 4331 documents. After removing the duplicates and irrelevant results, 114 articles were obtained for the complete revision of the full text. Following the final evaluation, 38 articles were included in the systematic review. The results are shown in three tables. Table 1 includes the general characteristics of the dogs. Table 2 contains information regarding the research goals and activities with the dogs. Table 3 provides the selection, training methods, and information about the health status of the dogs, including whether or not the dog belonged to the handler.

An overview of the results is presented in the following histogram chart (Figure 2).

### 3.1. General Information on Dogs

Two articles were published in 2021 [39,40], 9 studies were published in 2020 [41,49], 15 were published in 2019 [50,51,52,53,54,55,56,57,58,59,60,61,62,63,64], 3 were published in 2018 [65,66,67], 6 were published in 2017 [68,69,70,71,72,73], and 3 were published in 2016 [74,75,76]. Between 1 and 17 dogs were included in the studies. Nine studies did not specify the number of dogs [40,42,51,55,67,70,72,73,75]. The ages of the dogs ranged between 18 months and 13 years. Eight studies worked with dogs ranging from 1 to 5 years of age [41,43,44,50,55,57,59,61,66], and three studies worked with dogs 5 years or older [53,58,76]. Two studies, on the other hand, involved dogs with a wider age range (from 2 to 10 years [63], and 4, 6, and 8 years [65]). Two studies did not specify the age in years but only that the dogs were adults [47,72]. The remaining 20 articles did not indicate the dog’s age. Six studies included dogs of both sexes [39,47,55,59,61,65]. In two studies, all dogs were male [44,52], and in eight studies, all dogs were female [41,43,50,53,57,58,66,76]. The remaining 22 studies did not specify the sex of the dog. The breeds predominantly found in the results were Retrievers (Labrador, Golden, and Labradoodle), individually or together. The remaining studies worked with high breeds or crosses. Ten studies did not specify the breed of the dog [39,42,49,60,68,69,70,72,74,75].

### 3.2. Research Goals and Activities with Dog

In the AAT activity reports, almost all the works described the activities carried out with the dog; only one study did not release any information [40]. The sessions described had a duration between 10 min and 1 h. Activities included social [56], physical [39,55,59,60,62,66], and free [42,43,48,49,51,57,64,68,73] interactions; activities of rehabilitation [46,76]; psychomotor activities; and socialization [52]. Other activities included grooming, feeding, and stroking [54,58,61,67,70,75]; playing with the dog; and engaging in obedience exercises [45,50,65,72,74,75]. In one study, children read in the company of the dog [47]. Finally, in most studies, the role of the dog was focused on reducing anxiety and stress and improving mood. In four studies, the dogs worked with autistic patients [41,48,60], and in one study, the dogs also worked with Down-syndrome patients [52]. Four papers focused on the elderly to improve the physiological parameters of heart disease [66], Alzheimer’s disease [62], the perception of pain in geriatrics [59], and cognitive functions among patients with dementia [60]. In two papers, the dogs supported patients with oncological diseases [67,70]. Three studies carried out psychological rehabilitation [46] with inmates [72] and schizophrenia [74]. Other work focused on improving communication and social skills [44,45,47,49,51,63,76]. Finally, one study explored the use of support dogs during dental sessions [61].

### 3.3. Methods for Choosing the Dog, Dog Temperament, Dog Training, and Health Status (i.e., Behavioral Veterinary Medical Examination and Health Protocols) of Dogs

As a method for choosing the dog, five papers used test administration [41,44,51,65,72], while three papers evaluated AAT certifications [50,53,56]. Seven studies evaluated accredited programs as a method of inclusion [39,40,43,61,69,73,76]. All other studies did not include this information. To validate the dog’s temperament, most studies tested docility, socialization, or type of training. Thirty-two papers did not report any information on the temperament of the dogs. For training, almost all papers reported membership in accredited associations or programs as the source of dog training, but the type of training was rarely mentioned. Nine papers did not offer any information on the type of dog training [45,47,52,57,58,61,69,70,74]. Furthermore, dogs were examined by a behavioral veterinarian in only seven studies [41,44,63,67,72,74,76]. In terms of health protocols, almost all the reviewed works applied vaccinations and parasitic treatments. In 19 studies, no information on health protocols was provided.

Finally, in only three studies [55,70,76] was the dog handler also the owner. In most of the studies, whoever worked with the dog was only a handler, or it was not specified whether the handler was also the owner of the dog. The remaining 12 studies did not include this information.

## 4. Discussion

Dogs are some of the most frequently involved animals in AATs due to their long history of coevolution with humans and their numerous beneficial effects, which have been widely described in the literature [14,15,16].

However, although studies on dogs are numerous, many important aspects remain overlooked [20]. In most of the selected articles, there was often a lack of information regarding the general characteristics of the dog, such as the number of dogs involved in the AAT, as well as the age, sex, and breed of each dog. This is probably because other aspects of research are given importance, while the dogs have been overlooked while representing an integral and fundamental part of the AAT’s team [14,15,16]. In particular, data on the dog’s temperament are presented in very few studies [45,49,50,52,59,63,65,68,73], thereby neglecting the importance of who the dog is; in this way, the dog’s vision as the “other”, a unique subject with its own personality, is lost. The methods used to select dogs are also not exhaustively presented. Accredited methods of selection are rarely mentioned. Some works cited various tests carried out [44,51,65,76], others cited certifications [41,43,50,53,56,61], and others cited self-contained training programs [39,69]. This vast heterogeneity in choice protocols is linked to the lack of univocal tests to select a dog most suitable for a certain type of AAT and the overall lack of standardized selection criteria for the dogs involved. Furthermore, most of the works limited themselves to only mentioning the names of accredited associations or programs without describing the training methods of the dogs in detail. This result indicates a lack of unique educational training programs among the dogs involved.

In terms of the relationship between dog and handler [11,16,77], based on our results, the handler was reported to be the owner of the dog in very few studies [55,70,76], while most of the remaining studies did not specify whether the handler was also the owner of the dog. This is a very important aspect, as the relational dimension is essential for an effective intervention [16,77,78,79]. Furthermore, it was previously shown that the relational factors between dogs and owners influence the performance of the dog [78]. As other studies already reported, good attachment and a harmonious relationship between dog and owner can increase safety, reduce stress factors in the dog, and improve the dog’s performance [79]. Therefore, an interspecific relationship in an AAT can positively affect the final results of the intervention [80]. An equally important aspect that was neglected in the analyzed works concerns the applied health protocols. Few studies preventively examined dogs using a behavioral veterinarian, the only professional figure able to certify the suitability of a dog with respect to the absence of behavioral diseases [41,44,49,63,67,72,74,76]. This aspect is of fundamental importance for the safety of users involved in AAT to prevent zoonotic risks [81,82,83]. In addition, in some studies, the health protocols used to ensure the safety of the setting and the users/patients involved in the interventions were not mentioned or specified [40,42,43,45,47,48,49,50,52,55,56,57,59,60,64,65,66,71,74,75]. This result indicates the absence of standardized reference health protocols, which could yield dangerous consequences linked to zoonotic risks, especially since AATs are often given to immunosuppressed patients [84,85,86,87].

In terms of the types of activities carried out by the dogs during AATs, our results are in accordance with previous research findings [88], which highlighted the limitations of current studies due to a lack of information on the welfare indicators of dogs—particularly how different interventions (in terms of duration and frequency), the environmental factors of the setting, human manipulation, animal characteristics, etc. can influence these indicators. We also noted the physiological and behavioral variations that can occur in response to the actions of the handler or the patient, suggesting that we should measure the personality and stress levels not only of the dogs but also of the people involved, which represents a potential area of future research on interspecific relationships.

Therefore, we strongly suggest a greater awareness of scholars on the need to investigate on welfare, considering not only stress parameters, but also the pleasure indicators of dog during an AAT intervention.

In today’s society, dogs are recognized as sentient beings, as they are endowed with their own personalities [89,90] and emotions [91]. The data reported by Horowitz [89]—similar to the data reported in the present review—underline that in human–animal interactions, dogs are strongly neglected; however, since dogs are now considered to be sentient beings, their roles must be more seriously considered.

### Limitations

This study has several limitations. The heterogeneity of the data collected in this systematic review did not allow for adequate statistical evaluation of the different characteristics analyzed in each paper, as the examined data did not refer to universally shared protocols. Moreover, certain characteristics were not always presented or described. However, these data can be considered an important index of the heterogeneity that characterizes the scientific literature related to the characteristics and roles of dogs in AATs. Furthermore, these characteristics reinforce the need to develop and disseminate general protocols to facilitate the use of globally recognized tools and allow, at the same time, a greater focus on the dog.

## 5. Conclusions

Based on our results, the characteristics of the dogs involved in therapy are rarely considered, as such factors were neither described nor investigated in detail in most of the reviewed studies. With a contemporary and scientifically validated understanding of dogs as sentient beings, species characteristics and dog individuality must be seriously considered. Our results show that the parameters most requiring in-depth analyses are related to the choice, temperament, and training of the dog and health protocols. Furthermore, greater attention should be given to interspecific relationships and the factors that influence them, as well as the interspecific relational skills of each individually chosen animal and its relationship with the owner, such as attachment styles and relational reciprocity between the animals, patients, and members of the working group. All these factors will affect the animal’s welfare, the safety of the therapeutic setting, and achievement of the study’s goals. The strength of these therapies lies in the involvement of dogs, which offers its own languages and abilities that determine activation of the emotional, cognitive, and behavioral dynamics responsible for the various positive effects observed in previous studies. Therefore, it is necessary to implement standardized and univocal protocols that emphasize the role of the dog and facilitate a paradigm change capable of underscoring the importance of and respect for the individuality of each dog, not only the benefits the dog can bring.

## Figures and Tables

**Figure 1 animals-11-02576-f001:**
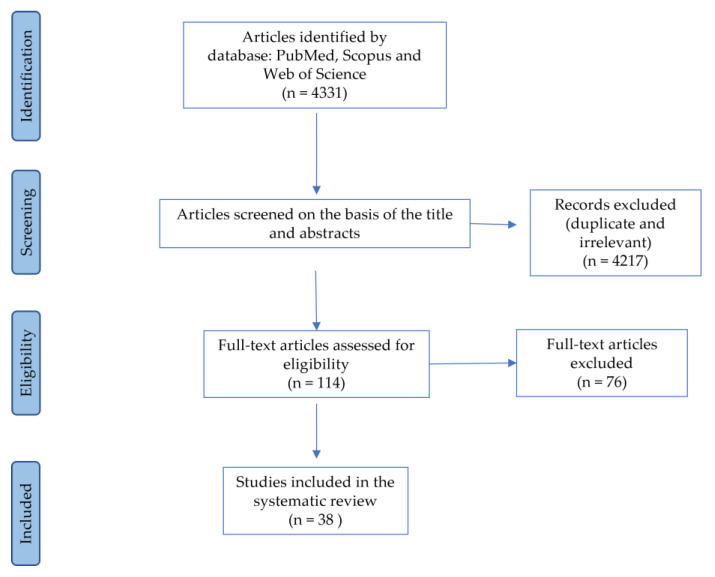
Flow diagram of the steps followed in the search strategy.

**Figure 2 animals-11-02576-f002:**
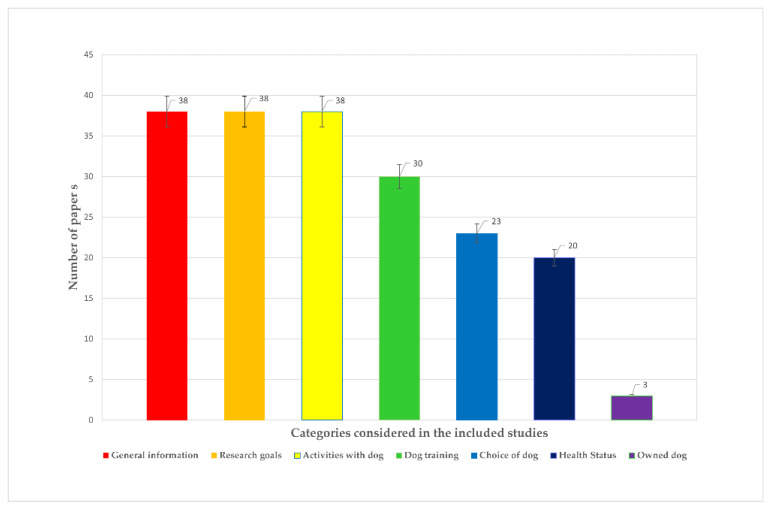
Overview of the results.

**Table 1 animals-11-02576-t001:** Results of general information regarding dogs involved in the included papers.

First Author, Year	General Information on Dogs				Reference
	No	Age (Years)	Sex	Breed	
Mueller M., 2021	4	8–13	m/f	n.a.	[39]
Allen B., 2021	n.a.	n.a.	n.a.	Labrador	[40]
Hill J., 2020	1	3	f	Labradoodle	[41]
Vincent A., 2020	n.a.	n.a.	n.a.	n.a.	[42]
Kashden J., 2020	1	2	f	poodle mix	[43]
Rodrigo-Claveron M., 2020	1	5	m	German shepherd	[44]
Jorgenson C., 2020	1	n.a.	n.a.	Labrador	[45]
Pruskowski K., 2020	6	n.a.	n.a.	Golden, Poodle, Labrador, Shetland sheepdogs, Collie	[46]
Rousseau C. X., 2020	5	adult	m/f	Bernese mountain dog, Maltese, Yorkshire terrier	[47]
Wijker C., 2020	4	n.a.	n.a.	Labrador crossbreeds, Poodles	[48]
Vidal R., 2020	2	n.a.	n.a.	n.a.	[49]
Machova K., 2019	1	5	f	Border Collie	[50]
Costa J., 2019	n.a.	n.a.	n.a.	Golden	[51]
Griffioen R., 2019	2	n.a.	m	Labrador, Labradoodle	[52]
Nilsson M., 2019	1	6	f	Labradoodle	[53]
Muela A., 2019	4	n.a.	n.a.	Labrador, Golden	[54]
Thompkins A., 2019	n.a.	>18 months	m/f	Golden, Terrier mix, Havanese, Labrador	[55]
Ambrosi C., 2019	6	n.a.	n.a.	golden, flat-coated	[56]
Perez-Saez E., 2019	1	3	f	Labrador	[57]
Perez M., 2019	1	10	f	Labrador	[58]
Rodrigo-Claverol M., 2019	3	4–3	m/f	Golden, Cavalier King Charles	[59]
Protopopova A., 2019	3	n.a.	n.a.	n.a.	[60]
Cruiz- Fierro N., 2019	4	average age of 40 months	m/f	English shepherd, Cchnauzer, Border Collie, Labrador	[61]
Sánchez-Valdeón L., 2019	1	n.a.	n.a.	Labrador	[62]
Wijker C., 2019	13	2–10	n.a.	Labradors, Labrador crossbreeds, Poodles, Golden, Golden crossbreeds, and a German Wirehaired Pointer.	[63]
Grajfoner, 2019	7	n.a.	n.a.	Labrador, lhasa apso, Cocker Spaniels, Golden, Collie-Spaniel, Border Collie	[64]
Dell C., 2018	3	4, 6, 8	m/f	Boxer and Bulldog	[65]
Handlin L., 2018	1	2	f	Labradoodle	[66]
Silva N.B., 2018	n.a.	n.a.	n.a.	Labrador, golden	[67]
Ward-Griffin E., 2017	7–12	n.a.	n.a.	n.a.	[68]
Binfet J., 2017	15–17	n.a.	n.a.	n.a.	[69]
Chubak J., 2017	n.a.	n.a.	n.a.	n.a.	[70]
Giuliani F., 2017	1	n.a.	n.a.	Border Collie	[71]
Contalbrigo L., 2017	n.a.	adult	n.a.	n.a.	[72]
Fiocco A., 2017	n.a.	n.a.	n.a.	Irish setter, Schnoodle, Miniature Poodles, Greyhound, King Charles Spaniel, Golden, and Australian Cattle Dog	[73]
Calvo P., 2016	5	n.a.	n.a.	n.a.	[74]
Swall A., 2016	n.a.	n.a.	n.a.	n.a.	[75]
Menna L.F., 2016	1	7	f	Labrador	[76]

Legend. f: female; m: male; n.a.: not available; min: minutes; h: hour.

**Table 2 animals-11-02576-t002:** Results of information on the “research goals” and “activities with the dog” in the included papers.

First Author, Year	Research Goals	Activities with Dog	Reference
Mueller M., 2021	reduce anxiety when experiencing a social stressor	social and physical interactions	[39]
Allen B., 2021	improve the severity of PTSD symptoms	n.a.	[40]
Hill J., 2020	achievement of occupational goals among autistic children	7 sessions for 1 h occupational activities	[41]
Vincent A., 2020	reducing anxiety and situational fear among children	1 h of free interactions	[42]
Kashden J., 2020	improve mood in an outpatient setting	20 weeks of sessions lasting 50 min Mutt-i-grees curriculum, which is based on the concepts of human–dog interactions	[43]
Rodrigo-Claveron M., 2020	improve communication and mobility of people with cognitive impairments	2 session per week for 6 months, physical interactions	[44]
Jorgenson C., 2020	increase verbal statements through various contingencies	3–7 sessions per day, 1–2 days a week (30 min/1 h) Playing with the dog	[45]
Pruskowski K., 2020	improving duration and quality of rehabilitation sessions and physical therapy	rehabilitation activities: walking with the dog; caregiving activities: brushing, petting	[46]
Rousseau C.X., 2019	supporting reading motivation among young children	45 min per session, children read in the company of the dog	[47]
Wijker C., 2020	improving the social development of adults with autism spectrum disorder	60 min session, interaction activities	[48]
Vidal R., 2019	improvements in social skills, a reduction in internalized and externalized symptomatology	weekly sessions of 45 min over 3 months, interaction activities.	[49]
Machova K., 2019	positively activate hospitalized patients	12 weeks of sessions lasting 20 min walks, playing ball, obedience exercises	[50]
Costa J., 2019	stuttering treatment	12 weeks of sessions lasting 50 min. Passive participation: the dog was wearing a vest containing pictures, words, and phrases. Active participation: selected an object. Participants were allowed to interact with the dog voluntarily during all sessions. Free interactions	[51]
Griffioen R., 2019	help children with autism-spectrum disorder and Down syndrome (behavioral synchrony)	6 weekly sessions of 30 min psychomotor and socialization activities, obstacle course	[52]
Nilsson M., 2019	complementary treatment in pediatric hospital care	a calm period and an active period with tricks guided by the handler	[53]
Muela A., 2019	reduce the internalizing and externalizing symptoms associated with traumatic stress disorder in children exposed to domestic violence	14 weeks of sessions lasting 1 h playing, carrying out tasks such as grooming and feeding, etc.	[54]
Thompkins A., 2019	improve affect, reduce stress, and reduce affect for red pain in individuals undergoing occupational therapy during rehabilitation from traumatic SCIs	physical interactions	[55]
Ambrosi C., 2019	effectiveness on depression and anxiety among institutionalized elderly individuals	10 sessions per week for 30 min of verbal and nonverbal social interactions, stroking the dog, giving or throwing food or a toy	[56]
Perez-Saez E., 2019	improving social behaviors and emotional expression among people with dementia	free interactions	[57]
Perez M., 2019	reducing anxiety in pediatric patients preparing for MRI	20–60 min of the interaction included sitting near the dog, petting the dog, and engaging in low-level play	[58]
Rodrigo-Claverol M., 2019	improving pain perception in polymedicated geriatric patients with chronic joint pain	12 weekly sessions of 60 min of physical interactions	[59]
Protopopova A., 2019	improving academic tasks in children with autism spectrum disorder	30 min session, once per day, 2–3 days per week, free interactions	[60]
Cruiz-Fierro N., 2019	help control anxiety during dental procedures	touching or stroking the dog during periods of stress	[61]
Sánchez-Valdeón L., 2019	improving the quality of life of people with Alzheimer’s disease	weekly session (1 h) for 12 months, physical interactions	[62]
Wijker C., 2019	reducing stress and improving social awareness and communication among adults with ASD	2 h per day (non-consecutive), maximum of 2 days per week	[63]
Griffioen R.E., 2019	improve students’ mood, well-being, and anxiety	1 session for 20 min handler and dog interactions or dog-only interactions	[64]
Dell C., 2018	improve welfare of prisoners	24 sessions over 8 months 30 min experiential learning with the dog, basic obedience (e.g., sit, heel)	[65]
Handlin L., 2018	improve systolic blood pressure/rate and heart disease among the elderly	physical activity (stroking, playing, etc.)	[66]
Silva N.B., 2018	improve physiological and psychosocial variables of pediatric oncology patients	30 min session per week caregiving activities, socialization, sensorial and upper limb stimulation	[67]
Ward-Griffin E., 2017	reducing pre-exam stress among students	90 min session, free interactions with the dogs	[68]
Binfet J., 2017	reduce stress among college students	drop-in program once a week, variable time, spending time with the dog	[69]
Chubak J., 2017	potential benefits for young people hospitalized with cancer	20 min of stroking the dog, the dog showing a trick to the patient	[70]
Giuliani F., 2017	benefits for individuals with difficulties learning	30 min playing ball, petting, and brushing the dog	[71]
Contalbrigo L., 2017	helping in the rehabilitation of drug-addicted prisoners	once a week for 6 months (60 min): cooperative games, problem solving games, agility dog path, role playing, obedience exercises, contact and care activities, olfactory discrimination games, exercises of communication with the body, free interactions	[72]
Fiocco A., 2017	buffering effect of stress responses among college students	10 min free interactions	[73]
Calvo P., 2016	rehabilitation for patients with schizophrenia	6 months of biweekly 1 h sessions, emotional bonding, dog walking, and dog training with play	[74]
Swall A., 2016	promote human welfare and stimulate training of physical, social, and cognitive functions.	the activities for the patient included close contact with the dog by touching its fur, cuddling, and talking, searching for hidden sweets, throwing balls, or other activities	[75]
Menna L.F., 2016	the therapeutic approach was based on the stimulation of cognitive functions such as attention, language skills, and spatial–temporal orientation.	intervention occurred once a week for 45 min over a 6-month period. Activities with the dog were based on the formal ROT protocol	[76]

Legend. n.a.: Not Available; SCI: Spinal Cord Injury; MRI: Magnetic Resonance Imaging; ASD: Autism Spectrum Disorder; ROT: Reality Orientation Therapy.

**Table 3 animals-11-02576-t003:** Results of information on “Choice of dog”, “Dog training”, “Health Status”, and “Dog ownership” in the included papers.

First Author, Year	Choice of Dog		Dog Training	Health Status		Dog Ownership	Reference
	Methods of Choosing	Temperament		Behavioral Veterinary Medical Examination	Health Protocols		
Mueller M., 2021	Pet Partners Program	n.a.	Pet Partners Program	n.a.	yes	n.a.	[39]
Allen B., 2021	retired service dogs	n.a.	local service dog organization	n.a.	n.a.	n.a.	[40]
Hill J., 2020	examination temperament and behavior	n.a.	obedience	yes	yes	handled	[41]
Vincent A., 2020	n.a.	n.a.	Therapy Dogs International or Pet Partners	n.a.	n.a.	handled	[42]
Kashden J., 2020	independently licensed therapy dog training facility	n.a.	12 weeks, standard therapy dog training	n.a.	n.a.	handled	[43]
Rodrigo-Claveron M., 2020	Liackhoff test	n.a.	clicker training	yes	yes	n.a.	[44]
Jorgenson C., 2020	n.a.	Assistance Dogs International	n.a.	n.a.	n.a.	handled	[45]
Pryskowski K., 2020	n.a.	n.a.	American Kennel Club canine good citizen. Therapy organization: Therapy Animals of San Antonio, Pet Partners, and Alliance of Therapy Dogs	n.a.	yes	handled	[46]
Rousseau C. X., 2020	n.a.	n.a.	n.a.	n.a.	n.a.	handled	[47]
Wijker C., 2020	n.a.	n.a.	Dutch Service dog Foundation	n.a.	n.a.	handled	[48]
Vidal R., 2020	n.a.	trained and tested to work with people	CTAC Method (Center of dog assisted Therapy)	yes	n.a.	n.a.	[49]
Machova K., 2019	AAT certification	interest in working with people, no aggressivity	obedience, to cope with stressful situations	n.a.	n.a.	handled	[50]
Costa J., 2019	tested for desirable reactions in responses to unknown people; unpredictable visual and sound stimuli; aggressive human voice; threatening gestures; places with a large concentration of people;	n.a.	Instituto Cão Terapeuta and Amor Canino Terapia organizations	n.a.	yes	handled	[51]
Griffioen R., 2019	n.a.	mild-mannered	n.a.	n.a.	n.a.	handled	[52]
Nilsson M., 2019	certificated for use with children in health care	n.a.	trained for use with children in health care	n.a.	yes	handled	[53]
Muela A., 2019	n.a.	n.a.	positive reinforcement techniques	n.a.	yes	n.a.	[54]
Thompkins A., 2019	n.a.	n.a.	Hand in Paw	n.a.	n.a.	owned	[55]
Ambrosi C., 2019	certification aptitude tests for therapy dogs	n.a.	professionally trained	n.a.	n.a.	handled	[56]
Perez-Saez E., 2019	n.a.	n.a.	n.a.	n.a.	n.a.	n.a.	[57]
Perez M., 2019	n.a.	n.a.	n.a.	n.a.	yes	handled	[58]
Rodrigo-Claverol M., 2019	n.a.	suitable character	Ilerkan Association	n.a.	n.a.	n.a.	[59]
Protopopova A., 2019	n.a.	n.a.	American Kennel Club Canine Good Citizen evaluation. All dogs are certified through and registered with a national therapy dog registry (e.g., Pet Partners, Alliance of Therapy Dogs)	n.a.	n.a.	handled	[60]
Cruiz- Fierro N., 2019	certification Council for Professional Dog Trainers (CCPDT)	n.a.	n.a.	n.a.	yes	handled	[61]
Sánchez-Valdeón L., 2019	n.a.	socialized, had a stable, friendly nature	trained for this purpose by a canine specialist	n.a.	yes	n.a.	[62]
Wijker C., 2019	n.a.	trained and tested to work with people	Dutch service dog foundation (guidelines to protect and monitor animal welfare)	yes	yes	n.a.	[63]
Grajfoner, 2019	n.a.	n.a.	therapet, Canine Concern Scotland Trust (Scottish Charity No)	n.a.	n.a.	handled	[64]
Dell C., 2018	screening, orientation, evaluation, and placement	mild-mannered, energetic, or laid-back	St. John’s Ambulance Therapy Dog program	n.a.	n.a.	handled	[65]
Handlin L., 2018	n.a.	n.a.	1 year, “Vårdhundskolan” (Sweden)	n.a.	n.a.	handled	[66]
Silva N.B., 2018	n.a.	n.a.	docility, obedience training, and socialization	yes	yes	handled	[67]
Ward-Griffin E., 2017	n.a.	no history of aggression or biting, good obedience, and friendly interactions with strangers	Vancouver ecoVillage Therapy Dog Programme	n.a.	yes	handled	[68]
Binfet J., 2017	BARK program	n.a.	n.a.	n.a.	yes	handled	[69]
Chubak J., 2017	n.a.	n.a.	n.a.	n.a.	yes	owned	[70]
Giuliani F., 2017	n.a.	n.a.	Swiss Romande Cynology Federation	n.a.	n.a.	n.a.	[71]
Contalbrigo L., 2017	simulation test	n.a.	specifically trained to perform do-assisted interventions with various kinds of patients; well-socialized	yes	yes	handled	[72]
Fiocco A., 2017	St. John’s Ambulance Therapy Dog program	docile manner	trained to interact with people in a docile manner	n.a.	yes	n.a.	[73]
Calvo P., 2016	n.a.	n.a.	n.a.	yes	n.a.	n.a.	[74]
Swall A., 2016	n.a.	n.a.	the dog is trained to know how to approach the person in a soft, gentle way (Swedish Standard Institute)	n.a.	n.a.	handled	[75]
Menna L.F., 2016	according to Federico II Model of Healthcare Zooanthropology	n.a.	educational program for pet therapy at the La Voce del Cane Dog Educational Centre, which follows the guidelines of the Italian National Educational Sports Center	yes	yes	owned	[76]

Legend. n.a.: Not Available.

## Data Availability

Data are available upon request to the corresponding author.

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
