# Peer review of "The Research of Standardized Protocols for Dog Involvement in Animal-Assisted Therapy: A Systematic Review"

_animals, 2021, doi:10.3390/ani11092576_

Round 1

Reviewer 1 Report

The new version of the paper is substantially improved as compared to the previous one and I would like to congratulate the authors for that. However, there are still some (minor) points that need to be addressed before the paper can be accepted for publication. I detail them in what follows.

Line 17: Do the authors actually mean “improve the animal’s welfare” or rather assess/evaluate?

Line 26 (and across the entire paper): Please use sex (biological), not gender (identity)!

Line 44: Please change “dogs” to “them”.

Lines 73-74: “The therapeutic benefits are (…) psychological (i.e., post-traumatic stress)” – post-traumatic stress is not a benefit, its treatment is. Please rephrase.

Line 77: Please provide definition for PTSD.

Line 81: Building a research question by stating that “in your opinion” there may be a relationship between variables must be avoided. Please rephrase and ideally present literature/references to back up your claims/hypothesis.

Line 98: Please provide definitions for AAA and AAE.

Line 121: 38 papers, not 39.

Figure 1: “irrelevant” is misspelled.

Figure 2: What is depicted here? Please add titles to the axes.

Line 158-159: The results on dog ownership are reported out of place (they should be reported in section 3.3 and not 3.1, since they are presented in Table 3).

Line 186: Please clarify in the text what is meant with the concept of health status. Is it behavior and physical health? This should help to understand the discussion part around this issue (see below).

Discussion (line numbers missing here):

This type of overstatements/moral statements should be avoided: “These lacunae underscore the lack of importance given to the dogs themselves, which are not considered an integral and fundamental part of the work AAT’s team”. I suggest that the authors adopt a parsimonious approach and state simply and objectively something like “dogs have been overlooked”. The present study did not aim at and no data was collected that allow any conclusion to be drawn regarding the reasons leading to this gap in the reporting of dog data. I had already raised concerns about this in my previous review, but I feel they are not yet fully acknowledged.

“This is a very important aspect, as the relational dimension is essential for an effective intervention.”: I think this statement requires a reference.

“An equally important aspect that was neglected in the analyzed works concerns the applied health protocols. Few studies preventively examined dogs using a behavioral veterinarian, the only professional figure able to certify the suitability of a dog [42,45,50,64,68,73,75,77]. This aspect is of fundamental importance for the safety of users involved in AAT to prevent zoonotic risks related to animal behavior [82,83,84]” – These ideas need rephrasing and further clarification. Namely:

  • What is the connection between applied health protocols and behavioral evaluation of dogs? (see comment above on the definition of health status)
  • Why state that vet behaviorists are “the only professional figure able to certify the suitability of a dog”? Wouldn’t well trained dog trainers be able do the job as well?
  • What are “zoonotic risks related to animal behavior”?

“In terms of the types of activities carried out by the dogs during AATs, we agree with the conclusions of previous works [89]” – Please rephrase. The authors should not write about whether they agree or not. Science is not about opinions; what is on the table is the fact that “the present results are in accordance with previous research findings”.

“which highlighted the limits of current studies” -- I think the authors mean limitations.

Reviewer 2 Report

The manuscript changes I requested have been appropriately edited and I feel the paper is very ready for publication. 

Author Response

We thank the Reviewer for his/her approval.

Reviewer 3 Report

None. All my comments appear to be addressed. Still needs some minor editing.

Author Response

Dear Reviewer, Thank you for your approval.

This manuscript is a resubmission of an earlier submission. The following is a list of the peer review reports and author responses from that submission.

Round 1

Reviewer 1 Report

The current study presents a systematic review of the Animal Assisted Therapy literature published between 2016 and 2021 aimed at identifying the dog data reported in these studies.

Although I think the goal of the paper is of value and that the presented results are worth being published, I will have to ask the authors to work on some core changes and then re‑submit the manuscript so that I can go on a second round of more detailed revisions. The major reason for this decision lays with the fact that the reading of the paper is very difficult, mainly due to poor English writing. I would advise the authors to seek the help of a native English speaker. In its current form, several parts of the paper are quite unintelligible and very exhausting for the reader to make sense out of what is being said. There are also several typos, lack of punctuation and abbreviations without definition across the paper. Please check them carefully.

Besides this major concern, I would like to advance a few more on which the authors can work right away before re-submitting:

- Explanation of materials and methods is confusing, it needs re-organization and more careful description of the steps of the review process (please check that figure and text do not match completely).

- Results: Figure 2 - What is depicted on it? Additionally, the results on dog ownership are reported out of place (they should be reported in section 3.3 and not 3.1, since they are presented in Table 3).

- Discussion: I would suggest a more focused/objective discussion. The authors seem to digress (and even going towards a moral perspective) sometimes. I will leave one example here: “In particular, the data concerning the dog's temperament are present in very few works [46,50,51,53,60,64,66,69,74], neglecting the importance of who the dog is, in this way its vision as otherness and as a unique subject with its own personality is lost” - The fact that the dog’s temperament in rarely presented does not necessarily mean that the authors are “neglecting the importance of who the dog is”, it may simple mean that not enough importance has been attributed to it (and here the authors need to state with clarity why this is of importance). Please revise the entire paper (and especially the Discussion) according to this suggestion.

In addition, some ideas also seem not well connected (especially, but not only, in the Discussion). The connection of ideas in the following paragraph, for example, is difficult to understand: “Analyzing the relationship between dog and handler [16,78,11], from the results obtained, there are very few studies in which the handler is also the owner of the dog [56,71,77], and most of the remaining studies do not specify whether the handler is also the owner of the dog. In our opinion, this is a very important aspect, inasmuch the relational dimension is essential for the effectiveness of an intervention. Furthermore, it has been shown that the relational factors between dogs and owners influence the performance of the dog. [79]).” Maybe it is just the word “Furthermore” that makes it unclear, hence we may once again be facing an English writing issue.

I believe I will have further suggestions, but for now I would rather work on them after a more intelligible version of the manuscript is submitted.

Author Response

Dear Reviewer,
as suggested by you, we have proceeded to revise the manuscript especially regarding the editing of the English language and style through the service provided by MDPI. We hope that in this form, the manuscript will be easier to read and understand.
We thank you for taking the time to read our paper and for your availability.
Kind regards.

Reviewer 2 Report

line 174: you refer to the participants as "people with dementia", but on line 174 you say "schizophrenics". Modify this to "people with schizophrenia" please.

Why was PubMed eliminated as a source when reviewing journal articles. You state that you eliminated this source, but no justification was given. Please add a sentence or two to explain this. 

Reviewer 3 Report

I reviewed the manuscript "The research of standardized protocols for dog involvement in Animal Assisted Therapy: A Systematic Review." Overall, this is a well organized meta-analysis of the AAT/AAI literature specifically investigating the data collected on the dogs in these studies.

It stands out to me how little we actually consider the age, sex, breed, and other characteristics of dogs in AAT. This alone makes the study worth publication. However, I do have some recommendations to improve the overall quality of the paper.

Abstract:

Line 21: Be cautious using acronyms in the abstract. Not everyone will be familiar with these. For that reason, I recommend spelling out PRISMA here, and again the first time it is introduced in text.

Materials and Methods:

Line 119-121: What was the Cohen's kappa of your interrater test? Or at minimum, what was the threshold below which you threw out studies? Include this information for your reader. It is not enough to just say you did it. We need the numbers, especially on a review.

Discussion:

Page 14, second paragraph (line numbers stopped): "The data reported by Horowiz..."

  1. Is this a typo? Should it be Horowitz?
  2. This warrants a citation. 

Overall Manuscript:

This paper is sufficient for publication once the above items are addressed. However, it is a bit difficult at times to determine the motivation behind this review. Was the concern about the high variability of dogs? Is the concern one of replicability or transparency in reporting because of the lack of data collected on dogs? Are you focused on improving welfare?

Tightening up the introduction and discussion around the motivation for doing this work and its relevance would make this manuscript more useful and likely lead to more citations.
